# A Tractable, Transferable, and Empirically Consistent Fibrous Biomaterial Model

**DOI:** 10.3390/polym14204437

**Published:** 2022-10-20

**Authors:** Nicholas Filla, Yiping Zhao, Xianqiao Wang

**Affiliations:** 1School of ECAM, College of Engineering, University of Georgia, Athens, GA 30602, USA; 2Department of Physics and Astronomy, University of Georgia, Athens, GA 30602, USA

**Keywords:** computer modeling, fiber orientation distribution, fiber network, pore size, tortuosity

## Abstract

Stochastic modeling is a useful approach for modeling fibrous materials that attempts to recreate fibrous materials’ structure using statistical data. However, several issues remain to be resolved in the stochastic modeling of fibrous materials—for example, estimating 3D fiber orientation distributions from 2D data, achieving the desired fiber tortuosity distributions, and dealing with fiber–fiber penetration. This work proposes innovative methods to (1) create a mapping from 2D fiber orientation data to 3D fiber orientation probability distributions, and vice versa; and (2) provide a means to select parameters de novo for random walks employing the popularized von Mises–Fisher distribution given that the desired tortuosity of the path is known. The proposed methods are incorporated alongside previously developed stochastic modeling techniques to simulate fiber network structures. First, fiber orientation distributions vary significantly depending on how a fibrous material is formed, and projection distortion affects the measurement of fiber orientation distributions when reported as 2D data such as histograms or polar plots. Relationships are developed to estimate 3D fiber orientation distributions from 2D data, accounting for projection distortion and the variety of orientation distributions observed in fibrous materials. We show that without correcting for projection distortion, fiber orientation distribution parameters could have errors of up to 100%. Second, in stochastic modeling, fiber tortuosity is usually treated with random walks, but no relationship is available for choosing random walk inputs to generate a desired fiber tortuosity. Relationships are also developed to relate the input parameters of von Mises–Fisher random walks to the expected tortuosity of the generated path—a necessary link to modeling fiber tortuosity distributions tractably and with empirical consistency. Using the developed relationships, we show that modeling of tortuous fibers from a distribution could be sped up by ~1200-fold and the uncertainty of selecting appropriate parameters could be eliminated. Third, randomly placing fibers in a simulation domain inevitably results in fiber–fiber penetration, and correcting this issue requires changes to the simulated fibrous material structure through non-penetration conditions. No thorough remedy can be offered here, but we statistically quantify the effects of enforcing non-penetration conditions on the fiber shape and orientation changes as well as the overall fibrous material model. This work offers tractable and transferable methods for treating fiber orientation and tortuosity that allow for empirical consistency in the stochastic modeling of fibrous materials.

## 1. Introduction

Numerical modeling is a valuable tool for investigating the properties of complex fibrous materials (FMs). For instance, numerical modelling has offered deeper knowledge regarding the mechanics of FMs embedded with stiff inclusions [1], the elasticity and plasticity of plant cell walls [2], the heat and momentum transfer between FMs and shear-thinning liquids [3], and the acoustic properties of FMs [4]. The drive to obtain more detailed insights and to understand the structure–property relationships of FMs naturally inspires researchers to develop more realistic FM models. Existing approaches to modeling FM structures include physics-driven approaches [5,6,7], random sequential adsorption [7,8,9,10,11,12], image reconstruction [13,14,15,16], tessellation [1,17,18,19,20], and stochastic modeling [4,21,22,23,24,25,26,27,28]. Among these methods, stochastic modeling has attracted attention for its utility in mimicking the real structure of FMs de novo. In stochastic modeling, all FM structural properties are drawn from various parameter distributions. Ideally, if those distributions are extracted from experimental measurements, the model should be empirically consistent. Despite the success of stochastic FM models, there remain challenges and issues that should be addressed to further their potential. Specifically, popularized stochastic methods for treating fiber orientation and tortuosity often lack tractability, transferability, or empirical consistency. 

Many empirical studies analyzing the fiber orientation distributions of FMs have been reported. These characterization studies have conclusively shown that fibrous materials exhibit a range of material symmetries, such as nearly isotropic [29,30,31,32,33], transverse isotropic [26,29,30,31,34,35,36,37,38,39], orthotropic [31,32,33,36,39,40,41,42,43,44,45], and hexagonal [13,31,36,40,43,45]. Furthermore, fiber orientations often do not possess symmetry and are distributed anisotropically [32,39,40,41,45]. Therefore, ideally, a stochastic FM model should have a means to recreate these material symmetries and to incorporate asymmetries. Although popular distributions such as those reported by Xia [46,47,48,49,50] and Schladitz [21,22,23,27,51] are undoubtedly useful for stochastic modeling and have been successfully used to mimic the orientation distributions of FMs, they lack transferability because they are specially formulated for specific symmetries. Fortunately, distributions that can model arbitrary material symmetries and asymmetries have been rigorously studied and employed in neuroscience to map fiber orientations within neurological tissue. Prominent distributions that possess a high level of transferability and empirical consistency include the Watson [52,53] and de la Vallee Poussin [52,53,54,55] distributions. These distributions possess three properties that allow them to be seamlessly employed to recreate arbitrary fiber orientation distributions: First, the distributions are antipodally symmetric; that is, the fiber orientation ${\hat{n}}_{f}$ is equivalent to the fiber orientation $-{\hat{n}}_{f}$—a desirable simplification when modeling fiber orientation distributions. Second, multiple instances of the distributions can be superimposed without algebraic difficulty, which is the critical property giving rise to their flexibility. Lastly, these distributions have been thoroughly studied and translated to a vector form (convenient for computational work), a polar/azimuthal angle form (convenient for data analysis), and a tensor form. Moakher and Basser have published a comprehensive review on these distributions, where all of these forms are thoroughly discussed and documented [52]. However, regardless of a distribution’s transferability, it cannot be directly used to accurately model 3D FM structures using 2D data extracted experimentally without correcting for projection distortion. Projection distortion occurs when 3D orientation distributions are projected onto a plane to be reported as a histogram or a polar plot. This is the same phenomenon that distorts the globe when projected onto a map. Without a relationship to correct for the projection distortion, fitting 2D orientation data with any probability distribution function (e.g., Xia, Schladitz, Watson, de la Vallee Poussin) does not yield proper parameters for recreating the 3D FM structure. Lastly, mitigating fiber penetration is often a necessary task in stochastic FM modeling, and may significantly affect the final fiber orientations, but the extent to which fiber orientations are disturbed during this process has not been rigorously quantified. 

In addition, the treatment of curvilinear fibers in stochastic FM modeling is currently incomplete. The complex shapes that a fiber traces through space are typically reduced to a scalar quantity such as tortuosity [56,57,58,59] or straightness [40,60,61,62]. Methods based on random walks and perturbations have been useful for stochastically modeling tortuous fibers [21,22,23,25,63,64]. However, these popular methods lack a connection between the input parameters of the random walk/perturbation and the tortuosity of the generated path. Without this relationship, attempting to model fiber tortuosity drawn from any distribution is a matter of trial and error. For this reason, the current treatments of fiber tortuosity in stochastic modeling are not tractable and, hence, cannot be empirically consistent until this issue is addressed. Moreover, as is an issue with fiber orientation, fiber tortuosity is subject to change when non-penetration conditions are enforced in FM modeling. The extent to which the non-penetration conditions affect the fiber tortuosity remains to be quantified. 

Here, pertinent issues in stochastic FM modeling are addressed. First, the issue of projection distortion is addressed by developing a novel relationship to map from 2D fiber orientation data to 3D de la Vallee Poussin orientation distributions. The corrections from this mapping are shown to reduce parameter errors by up to 100%. Second, a novel empirical relationship is developed between the von Mises–Fisher random walk and the tortuosity of the path that it generates. The relationships for selecting von Mises–Fisher random walk parameters are shown to speed up the simulation of tortuous fibers by ~1200-fold and eliminate the issue of estimating parameters. Third, the effects of enforcing non-penetration conditions on fiber length, orientation, and tortuosity distribution are reported. Furthermore, the pore size of the generated FMs resulting from this model is measured and validated against known empirical relationships. Consequently, multivariate predictors for the mean pore size as a function of the FM’s structural parameters are revealed. The results are consistent with the known inverse power law for mean pore size versus volume fraction and the linear scaling of mean pore size with mean fiber diameter [20,63,65,66,67,68]. 

## 2. Methods

As discussed in Section 1, fibrous material structures can be modelled using a handful of random processes to reproduce the distribution of fiber diameter, length, orientation, and tortuosity observed in real structures. Here, the methods for modelling fiber diameter, length, orientation, and tortuosity are detailed. The methods employed to enforce non-penetration conditions and measure the pore size of the resulting models are also discussed.

### 2.1. Model Domain, Volume Fraction, and Spatial Distribution of Fibers

Modeling began by declaring a spatial domain Ω, the fiber volume fraction $\phi_{f}$, mean fiber length $\langle l_{f}\rangle$, and mean fiber diameter $\langle d_{f}\rangle$. The spatial domain was a box whose faces met at right angles with side lengths $L_{\Omega }$ and volume $V_{\Omega }=L_{\Omega }^{3}$. The number of fibers, $N_{f}$, was estimated as $N_{f}\approx \frac{4V_{\Omega }\phi_{f}}{\pi \langle l_{f}\rangle {\langle d_{f}\rangle }^{2}}$, and the result was rounded to the nearest whole number. Then, $N_{f}$ seeds, $x_{0}$, were uniformly distributed in the domain $x_{0,i}=L_{\Omega }\left[N_{ur,1},\;N_{ur,2},\;N_{ur,3}\right]$, where $N_{ur,i}$ is a uniformly random number between 0 and 1.

### 2.2. Drawing Random Numbers from Probability Distributions

In FM stochastic modeling, the geometric parameters of individual fibers—such as fiber orientation and tortuosity—are randomly drawn from distributions. Random numbers were drawn from a continuous probability distribution p(X) or histogram data H(X) using rejection sampling, where X is a randomly distributed variable. First, a number was uniformly sampled between [a, b] using $X^{\prime }=\left(b-a\right)N_{ur}+a$, where $N_{ur}$ is a uniformly random number between 0 and 1, $X^{\prime }$ is a sample value, and a and b are the lower and upper bound of the sampling domain, respectively. A second number, $p^{\prime }$, was uniformly sampled between [0,max(p(X))] for probability densities or [0,max(H(X))] for histograms. If $p^{\prime }<p\left(X^{\prime }\right)$, the sample value $X^{\prime }$ was accepted, and if $p^{\prime }>p\left(X^{\prime }\right)$, the sample value $X^{\prime }$ was rejected. When using histogram data, $H\left(X^{\prime }\right)$ was calculated by linear interpolation. For continuous probability distributions with infinite support, a was taken as μ−6σ and b was taken as μ+6σ, where μ and σ are the mean and standard deviation of the distribution, respectively. For continuous probability distributions with lower or upper finite support, a was set equal to the lower support and b was set equal to the upper support. For histogram data, a was taken as the left edge of the leftmost bin, and b was taken as the right edge of the rightmost bin. For 3D directional distributions, the same algorithm applied except that the distributed variable was a unit vector sampled from the unit sphere. 

### 2.3. Fiber Diameters and Lengths

In our fiber model, the fiber diameters had a mean $\langle d_{f}\rangle$ and a standard deviation $std\left(d_{f}\right)$, while the fiber lengths had a mean $\langle l_{f}\rangle$ and a standard deviation $std\left(l_{f}\right)$. Both fiber lengths and diameters were drawn from a gamma distribution:(1)$$p_{\Gamma }\left(X\right)=\frac{X^{k_{\Gamma }-1}e^{-X/\theta_{\Gamma }}}{\Gamma \left(k_{\Gamma }\right)\theta^{k}}$$
where $p_{\Gamma }\left(X\right)$ is the probability density for a fiber with diameter or length X, $k_{\Gamma }$ is the shape parameter, $\theta_{\Gamma }$ is the scale parameter, and $\Gamma \left(k_{\Gamma }\right)$ is the gamma function evaluated at $k_{\Gamma }$. The gamma distribution has a lower finite support at X=0 and an infinite upper support. The parameters for the gamma distribution were calculated from the mean and standard deviation of X using the following equations:(2)$$k_{\Gamma }=\frac{{\langle X\rangle }^{2}}{std{\left(X\right)}^{2}}$$
(3)$$\theta_{\Gamma }=\frac{std{\left(X\right)}^{2}}{\langle X\rangle }$$

### 2.4. Fiber Orientations

If a fiber is treated as a straight line, then a vector tangent to that line is defined as the fiber orientation. For a curvilinear fiber, the average of all tangent vectors along its backbone defines its mean orientation. A distribution of fiber orientations shows the material’s symmetry, such as transverse isotropic (Figure 1(a.i)), orthotropic (Figure 1(a.ii)), or hexagonal (Figure 1(a.iii)). Here, fiber orientations were drawn from the de la Vallee Poussin (VP) probability distribution: (4)$$p_{VP}\left({\hat{n}}_{f}\right)=\frac{2k_{VP}^{3D}+1}{4\pi }{\left({\left({\hat{n}}_{f}\cdot {\hat{m}}_{VP}\right)}^{2}\right)}^{k_{VP}^{3D}}$$
where $p_{VP}\left({\hat{n}}_{f}\right)$ is the probability density for a fiber (Figure 1(a.i–a.iv)) with orientation ${\hat{n}}_{f}$, unit vector ${\hat{m}}_{VP}$ is the preferred or mean orientation, and $k_{VP}^{3D}$ is the VP orientation strength parameter. Equation (4) is applicable to describe transverse isotropic fiber orientation distributions when $k_{VP}^{3D}>0$ or isotropic fiber orientation distributions when $k_{VP}^{3D}$ = 0. Any arbitrary orientation distribution can be treated as a linear combination of multiple VP distributions of different kernels:(5)$$p_{VP}\left({\hat{n}}_{f}\right)=\overset{N_{VP}}{\underset{i=1}{\sum }}\frac{2k_{VP,i}^{3D}+1}{4\pi N_{VP}}{\left({\left({\hat{n}}_{f}\cdot {\hat{m}}_{VP,i}\right)}^{2}\right)}^{k_{VP,i}^{3D}}$$
where $N_{VP}$ is the number of superimposed VP kernels. For isotropic and transverse isotropic orientation distributions $N_{VP}=1$; orthotropic $N_{VP}=2$; hexagonal $N_{VP}=3$; and asymmetric $N_{VP}$ can be arbitrary.

Equations (4) and (5) were fitted to 2D experimental histogram data using the polar (θ) and azimuthal (ϕ) angle forms of the VP distribution. To do so, let

${\hat{n}}_{f}=\left(cos\theta sin\phi ,\;sin\theta sin\phi ,\;cos\phi \right)$ and ${\hat{m}}_{VP,i}=\left(cos\theta_{VP,i}sin\phi_{VP,i},\;sin\theta_{VP,i}sin\phi_{VP,i},\;cos\phi_{VP,i}\right)$ with $0<\theta ,\;\theta_{VP}<2\pi$ and $0<\phi ,\;\phi_{VP}<2\pi$. The mathematics can be simplified by setting $\phi =\phi_{VP}=\pi /2$, and then Equations (4) and (5) can be rewritten as follows:
(6)$$p_{VP}\left(\theta \right)=\frac{k_{VP}^{2D}\Gamma \left(k_{VP}^{2D}\right)}{\sqrt{\pi }\Gamma \left(k_{VP}^{2D}+\frac{1}{2}\right)}{\left({\left(cos\theta cos\theta_{VP}+sin\theta sin\theta_{VP}\right)}^{2}\right)}^{k_{VP}^{2D}}$$
(7)$$p_{VP}\left(\theta \right)=\overset{N_{VP}}{\underset{i}{\sum }}\frac{k_{VP}^{2D}\Gamma \left(k_{VP,i}^{2D}\right)}{\sqrt{\pi }\Gamma \left(k_{VP,i}^{2D}+1/2\right)N_{VP}}{\left({\left(cos\theta cos\theta_{VP,i}+sin\theta sin\theta_{VP,i}\right)}^{2}\right)}^{k_{VP,i}^{2D}}$$

### 2.5. Fiber Tortuosity

The idealized picture of a fiber is a straight cylinder; a slightly more realistic description is a curvilinear cylinder. In practice, there are many measurements to define the curviness of a fiber or a path, such as straightness, sinuosity, fractal dimension, or tortuosity, with tortuosity being one of the most popular. Tortuosity, $\tau_{f}$, is the ratio of a curve’s length to the total distance traveled and is always greater than or equal to one. In this paper, fiber tortuosity $\tau_{f}$ was drawn from probability distributions or interpolated histograms and followed a lognormal distribution: (8)$$p_{ln}\left(\tau_{f}\right)=\frac{1}{\tau_{f}s_{ln}\sqrt{2\pi }}\exp\left(-\frac{{\left(\ln\tau_{f}-m_{ln}\right)}^{2}}{2s_{ln}^{2}}\right)$$
where $p_{ln}\left(\tau_{f}\right)$ is the probability density of a fiber having a tortuosity $\tau_{f}$, while $m_{ln}$ and $s_{ln}$ are the expected value and standard deviation of $\tau_{f}$’s natural logarithm, respectively. Both $m_{ln}$ and $s_{ln}$ can be calculated from the mean and standard deviation of $\tau_{f}$, as follows:(9)$$m_{ln}=\ln\left(\frac{\langle \tau_{f}\rangle {}^{2}}{\sqrt{std{\left(\tau_{f}\right)}^{2}+{\langle \tau_{f}\rangle }^{2}}}\right)$$
(10)$$s_{ln}=\sqrt{\ln\left(\frac{std{\left(\tau_{f}\right)}^{2}}{{\langle \tau_{f}\rangle }^{2}}+1\right)}$$

Tortuosity can never be less than 1, so Equation (8) is truncated at $\tau_{f}=1$, meaning that Equations (9) and (10) are only approximations here. 

Curves with tortuosity $\tau_{f,i}$ were generated using the von Mises–Fisher (VMF) directional probability distribution: (11)$$p_{VMF}\left({\hat{n}}_{VMF}\right)=\frac{\kappa_{VMF}}{4\pi \sinh\kappa_{VMF}}\exp\left(\kappa_{VMF}{\hat{n}}_{f}\cdot {\hat{n}}_{VMF}\right)$$
to guide random walks that were subsequently smoothed with cubic splines (Figure 1(b.i,bii)). Here, $p_{VMF}\left({\hat{n}}_{VMF}\right)$ is the probability of stepping in direction ${\hat{n}}_{VMF}$, ${\hat{n}}_{f}$ is the fiber orientation, and $\kappa_{VMF}$ is the VMF strength parameter. A fiber was generated from a smoothed VMF random walk with step size $\Delta_{VMF}$, spline step size $\Delta_{CS}$, and preferred direction ${\hat{m}}_{VMF}={\hat{n}}_{f}$ (Figure 1(b.i)). For a low orientation strength $\kappa_{VMF}$, the simulated fibers were highly tortuous (Figure 1(b.ii)), while for high orientation strength $\kappa_{VMF}$ they were less tortuous (Figure 1(b.ii)). Random walks began at fiber seeds $x_{0}$; a direction was drawn from the VMF distribution (Equation (11)), and a step was taken using $x_{i}=x_{i-1}+\Delta_{VMF}{\hat{n}}_{VMF}$, where $\Delta_{VMF}$ is the step size. The number of steps taken was calculated as $N_{VMF}=l_{f}/\Delta_{VMF}$. Afterwards, each random walk was smoothed with cubic spline interpolation with step size $\Delta_{CS}$. The criteria for choosing $\kappa_{VMF}$, $N_{VMF}$, $\Delta_{VMF}$, and $\Delta_{CS}$ are detailed in Section 3.2. 

### 2.6. A Fiber as a Chain of Particles and Non-Penetration Conditions

After employing the methods discussed in Section 2.1, Section 2.2, Section 2.3, Section 2.4 and Section 2.5, the result was a box full of randomly oriented, randomly positioned fibers. In this case, it is practically inevitable that some or many fibers are penetrating one another. This penetration needs to be addressed in many cases, especially when simulating non-rigid FM models. Here, classical mechanics were used to mitigate fiber–fiber penetration by introducing a repulsive force between fibers. In attempt to maintain the prescribed fiber tortuosity and length, restoring forces for bond lengths and angles were included between particles in a fiber. Lastly, the non-penetrating configuration is always a lower energy state than the penetrating configuration. A drag force was introduced to equilibrate the system towards the lower-energy non-penetrating configuration. 

A single fiber was represented as a chain of spherical particles ${\tilde{x}}_{f}$, whose positions were defined by the smoothed VMF walk. Periodic boundary conditions were used to reinsert particles initially existing outside the spatial domain. Harmonic bonds, cosine angles, and soft contact potentials were used to enforce non-penetration conditions (Figure 1(c.i–c.iii)), as follows:(12)$$f_{b,i}=-2k_{b}\left(R_{ij}-R_{0,ij}\right)\frac{r_{ij}}{R_{ij}}$$
(13)$$f_{\theta ,i}=k_{\theta }\sin\left(\theta_{ijk}-\theta_{0,ijk}\right)\frac{\left(r_{ij}\times r_{kj}\right)\times r_{ij}}{\left\|(r_{ij}\times r_{kj}\right)\times r_{ij}\|}\;$$
(14)$$f_{\theta ,k}=k_{\theta }\sin\left(\theta_{ijk}-\theta_{0,ijk}\right)\frac{\left(r_{kj}\times r_{ij}\right)\times r_{kj}}{\left\|\left(r_{kj}\times r_{ij}\right)\times r_{kj}\right\|}\;$$
(15)$$f_{c,i}=\frac{\pi k_{c}}{R_{c,ij}}\sin\left(\frac{\pi R_{ij}}{R_{c,ij}}\right)\frac{r_{ij}}{R_{ij}}$$
(16)$$f_{b,j}=-f_{b,i},\;f_{\theta ,j}=-f_{\theta ,i}-f_{\theta ,k},\;f_{c,j}=-f_{c,i}$$
where $f_{b}$, $f_{\theta }$, and $f_{c}$ are the forces felt by particles due to the bond, angle, and contact potential, respectively; $k_{b}$, $k_{\theta }$, and *k_c_* are the stiffnesses of the bond, angle, and contact potential, respectively; $r_{ij}$ is the distance vector between particles i and j; $R_{ij}$ is the distance between particles i and j; $R_{0,ij}$ is the equilibrium distance between particles i and j; $R_{c,ij}$ is the cutoff distance of the contact potential; $\theta_{ijk}=\mathrm{acos}\left(r_{ij}\cdot r_{kj}/R_{ij}R_{kj}\right)$ is the angle between particles i, j, and *k*; and $\theta_{0,ijk}$ is the equilibrium angle between particles i, j, and k. The equilibrium distance between two bonded particles in the fiber was taken as $\Delta_{CS}$ of the smoothed VMF walk. The equilibrium angle between three consecutive particles was taken as the initial angle between them after the smoothed VMF walk. The contact potential cutoff distance between particles i and j was taken as $\frac{d_{f,i}+d_{f,j}}{2}$. The contact potential was not calculated between bonded particles or particles participating in a bond trio.

Non-penetration conditions were enforced by running damped classical mechanics simulations. A drag force was added to each particle using Stoke’s law (Figure 1(c.iv)).
(17)$$f_{s}=-3\pi \eta d_{f,i}v_{i}$$
where η is the viscosity of a surrounding fluid and $v_{i}$ is the velocity of the particle. Using the forces defined in Equations (12)–(16), the velocities v and positions x of the particles were updated by the velocity Verlet time-stepping algorithm. The time step Δt was taken as $\frac{1}{50\sqrt{k_{b}/m}}\;$. Periodic boundary conditions were employed during the dynamics simulation, and fiber ends were held fixed for the first 10,000 timesteps to restrict fiber rotations during high-energy dynamics. 

The fiber elastic modulus E, Poisson’s ratio ν, particle mass m, and fluid viscosity η were chosen randomly, and linear elasticity theory was used to relate E to $\left\{k_{b},\;k_{\theta },\;k_{c}\right\}$. Bond stiffness, $k_{b}$, was set equal to $EA/R_{ij,0}$, where $A=\frac{\pi }{4}d_{f}^{2}$ is the cross-sectional area of the fiber. Angle stiffness, $k_{\theta }$, was set equal to $EI/R_{ij,0}$, where $I=\frac{\pi }{32}d_{f}^{4}$ is the second moment of area of the fiber. Contact stiffness, $k_{c}$, was set equal to $R^{9/4}\nu^{1/3}E/16$, where $1/R=1/d_{f,i}+1/d_{f,j}$ (approximately close to Hertz contact). Hertzian and other contact theories were applicable; however, the soft potential in Equation (15) was chosen to avoid large accelerations when the initial separation of two particles was much smaller than the sum of their radii.

After constructing the FM networks (Figure 1d), their corresponding structure parameters (such as fiber volume fraction, orientation, length, tortuosity, etc.) were characterized and compared to the distributions initially used for each simulation. 

### 2.7. Pore Size Measurement

Once a simulated network was finished, a watershed method [19,20,69] was used to calculate the pore diameters, $d_{p}$ (Figure 1e). First, the simulation domain was segmented into voxels (Figure 1e, top left). The voxel edge lengths were 5 times smaller than the smallest fiber diameter. If a voxel was contained or partially contained in a fiber particle, the voxel was given a value of 1; otherwise, it was given a value of 0 (Figure 1e, top right). The distance transform was applied to the voxels, where each voxel was assigned a value corresponding to the minimum distance from itself to a voxel with a value of 1 (Figure 1e, bottom left). By convention, the distance transform was negated, and the voxels located in local minima were marked as pores (Figure 1e, bottom right). Local minima were found using the watershed method. The pore diameters were taken as the value of the distance transform at local minima. 

## 3. Results and Discussions

### 3.1. Relating de la Vallee Poussin Parameters Measured from 2D data to 3D Structures

Fiber orientation is the mean direction that a fiber points towards, calculated by averaging the tangent vectors along the axial direction of a fiber. The distribution of fiber orientations is typically modelled with a 3D distribution but measured and reported as a 2D distribution. In Figure 2a, two transverse isotropic VP distributions with the same orientation strength are shown ($k_{VP}^{3D}$ = 6). The blue VP distribution lies in the projection plane, while the orange VP distribution has been rotated out of the “projection plane” by an angle ψ=45°. An orientation that could have been drawn from either distribution is shown as ${\hat{n}}_{f}$. Since experimentally the observer is measuring the orientation from above (i.e., projecting the 3D network into a 2D network), the effective orientation ${\hat{n}}_{f}^{\prime }$ being measured is the projection of ${\hat{n}}_{f}$ on the projection plane. Such a projection process could result in projection distortion, i.e., the observed 2D orientation distribution would vary when the ψ is changed with respect to the observer even though the 3D distribution shape remains the same. For example, two fiber orientation distributions with the same 3D fiber orientation strength ($k_{VP}^{3D}=6$) were generated according to the blue and orange distributions in Figure 2a, and the corresponding 2D orientation distributions on the horizontal projection plane were obtained. Figure 2b shows the corresponding orientation distribution histograms, and clearly the distribution for ψ=0° (blue histogram) is significantly different from that for ψ=45° (red histogram). Equation (6) was used to fit these two histograms, and the orientation strength parameter $k_{VP}^{2D}$ was extracted to be $k_{VP}^{2D}=15$ for ψ=0° and $k_{VP}^{2D}=8$ for ψ=45°. Note that the true 3D orientation strength $k_{VP}^{3D}=6$ is the same for both cases. Hence, without understanding how the projected 2D measurement affects the fiber orientation distribution, the “true” distribution cannot be recovered from 2D data.

In order to obtain the relationship of the 2D and 3D orientation distribution, a detailed investigation was conducted by systematically varying both $k_{VP}^{3D}\;\left(\mathrm{from}\;0\;\mathrm{to}\;10\right)$ and ψ (between 0° and 90°). Figure 2c plots the extracted $k_{VP}^{2D}$ as a function of $k_{VP}^{3D}$ for different ψ for transverse isotropic distribution. At a fixed ψ, the measured $k_{VP}^{2D}$ increases linearly with $k_{VP}^{3D}$, while the slope of the $k_{VP}^{2D}\;\mathrm{vs}.\;k_{VP}^{3D}$ plot changes as a function of $\cos^{2}\psi$ (Figure 2d; filled circles are data obtained from Figure 2c, and the solid curve is the fitting). Similar measurements and analyses were conducted for orthotropic and hexagonal distributions and showed identical trends. In fact, our systematic investigation revealed the following relationship: (18)$$k_{VP}^{3D}=\frac{3}{8}\frac{\sqrt{N_{VP}}}{\cos^{2}\psi }k_{VP}^{2D}$$

According to Equation (18), if one knows the relative orientation between the 3D orientation distribution and the projection plane of the measurement, one should be able to extract the true 3D orientation strength $k_{VP}^{3D}$ based on the measured $k_{VP}^{2D}$ value. Alternatively, if experimentally multiple orientation samplings were obtained, then:
(19)$$\begin{array}{cc} \bar{k_{VP}^{2D}}=\frac{1}{\Psi_{2}-\Psi_{1}} & \int_{\Psi_{1}}^{\Psi_{2}}\frac{8}{3}\frac{k_{VP}^{3D}}{\sqrt{N_{VP}}}\cos^{2}\psi d\psi \\ & =\frac{4k_{VP}^{3D}\left(\Psi_{1}-\Psi_{2}+\cos\left(\Psi_{1}\right)\sin\left(\Psi_{1}\right)-\cos\left(\Psi_{2}\right)\sin\left(\Psi_{2}\right)\right)}{\left(\Psi_{1}-\Psi_{2}\right)\sqrt{N_{VP}}}\; \end{array}$$

Thus, the true $k_{VP}^{3D}$ can be obtained experimentally by measuring the average $\bar{k_{VP}^{2D}}$ at different ψ orientation sampling. If the limits of integration span an integer multiple of π $\left(\Psi_{2}-\Psi_{1}=n\pi \right)$, then Equation (19) can be reduced to $\bar{k_{VP}^{2D}}=\frac{4k_{VP}^{3D}}{3\sqrt{N_{VP}}}$.

To further illustrate the relationship (Equation (18)) between the 3D and 2D orientation distributions, we simulated the FMs fibrin [29], stone wool [13], cross-laid polyester [43], and collagen [41] using experimental orientation data. The discrete data shown in Figure 2(e.i–h.i) are the experimentally measured 2D fiber orientation distributions. These data were fitted by the 2D VP distribution (Equation (7)); see the solid blue curve in each figure), and the corresponding fitting parameters are summarized in Table 1. 

For fibrin, stone wool, cross-laid polyester, and collagen, we obtained $k_{VP}^{2D}=0.98,\;\;k_{VP,1}^{2D}=k_{VP,2}^{2D}=k_{VP,3}^{2D}=5.25$, $k_{VP,1}^{2D}=k_{VP,2}^{2D}=2.5,$ and $[k_{VP,1}^{2D},k_{VP,2}^{2D}]=\left[4,\;2.2\right]$, respectively. Since experimentally no ψ-dependent measurements were conducted, we assumed that that ψ=0° for all four cases. Thus, based on Equation (18), the corresponding 3D orientation strength $k_{VP}^{3D}$ can be estimated as shown in Table 1. Visually it is quite difficult to tell the difference between the four FMs. The projected 2D orientation distributions were obtained from the simulated FMs and are plotted as orange histograms in Figure 2(e.i–h.i). Visually, the 2D orientation distributions obtained from the simulated FMs match quite well with the experimental distributions, with a slightly larger variation for collagen (Figure 2(h.i)). Moreover, as a comparison, when we assumed that $k_{VP}^{3D}=k_{VP}^{2D}$ (which has been often done in experiments, and we call it non-corrected situation), another set of FM models were generated for the above four cases, and the corresponding 2D orientations are plotted as the white histograms in Figure 2(e.i–h.i). Compared to the corrected projection distortion situation—i.e., the orange histograms—the non-corrected situation generally overestimates $k_{VP}^{3D}$. Therefore, the experimentally measured 2D orientation distribution parameters cannot be directly taken as parameters for 3D orientation distributions as usual, because this can lead to errors of up to 100% when estimating $k_{VP}^{3D}$. A good approximation is to use Equation (18) to convert the 2D parameters to 3D parameters. Based on both the experimental and fitting results for the 2D orientation distributions, the orientation distributions of fibrin, stone wool, and cross-laid polyester are quite symmetric, while that of collagen is not symmetric. Clearly, even with the distortion correction, the obtained $\left[k_{VP,1}^{2D},k_{VP,2}^{2D}\right]$ from the simulations does not match well with those from the experimental data of the asymmetric case. 

### 3.2. Relating $\kappa_{VMF}$ and $N_{VMF}$ to Fiber Tortuosity $\tau_{f}$

Fiber tortuosity is the ratio of a fiber’s contour length to its end-to-end length (Figure 3a). Controlling fiber tortuosity can be a tricky task in simulations because it is difficult to know the end-to-end distance of randomly generated curves in advance. This section details the analysis of random walks guided by the VMF probability density and smoothed with a cubic spline. 

The VMF random walk has four independent variables: the orientation strength parameter $\kappa_{VMF}$, the number of random steps $N_{VMF}$, the step size $\Delta_{VMF}$, and the separation of particles in the cubic spline $\Delta_{CS}$. The length of the smoothed random walk is approximately equal to $N_{VMF}\Delta_{VFM}$. Random walks with orientation strengths between 0.001 and 300, a number of steps between 3 and 80, step sizes between $\frac{l_{f}}{3}$ and $\frac{l_{f}}{80}$, and particle separations between $\frac{l_{f}}{3}\;\mathrm{and}\;\frac{l_{f}}{800}$ were used to generate 5800 fibers to study the relationships between smoothed random walk parameters and fiber tortuosity. Figure 3a shows three representative fibers with different $\tau_{f}$; when $\tau_{f}$ increases from 1.05 to 1.35, the fiber changes from a fairly straight fiber to a curved fiber with multiple bends. Preliminary analysis showed that $\Delta_{CS}$ had little-to-no effect on the tortuosity of a path generated by the VMF random walk. However, $\kappa_{VMF}$ strongly influenced the tortuosity of the resulting path, and $N_{VMF}$ limited the tortuosity that a random walk could achieve. The variables $N_{VMF}$ and $\Delta_{VMF}$ were correlated with one another; thus, only one of them needed to be studied here. The number of VMF steps was chosen as a control variable for the response variable $\tau_{f}$. In Figure 3b, the control variable $\kappa_{VMF}$ is plotted on the y-axis (log), and the dependent variable—the expectation value of tortuosity $\langle \tau_{f}\rangle$—is plotted on the x-axis, as is customary when inverting a problem. As is already known, the expected fiber tortuosity $\langle \tau_{f}\rangle$ was inversely proportional to the orientation strength parameter $\kappa_{VMF}$. However, decreasing $\kappa_{VMF}$ did not necessarily increase $\langle \tau_{f}\rangle$. If the number of VMF steps $N_{VMF}$ was too small, decreasing $\kappa_{VMF}$ led to no change in $\langle \tau_{f}\rangle$. This can be observed in Figure 3b,c (solid black curves); as $\kappa_{VMF}$ decreases, $\langle \tau_{f}\rangle$ approaches a limit value, $\langle \tau_{f}\rangle {}_{limit}$, for a fixed value of $N_{VMF}$. The orientation strength parameter was found as a function of the expected fiber tortuosity:(20)$$\kappa_{VMF}\left(\langle \tau_{f}\rangle \right)=\exp\left(\frac{-0.2{\langle \tau_{f}\rangle }^{3}-0.15{\langle \tau_{f}\rangle }^{2}+1.56\langle \tau_{f}\rangle -1.2}{{\langle \tau_{f}\rangle }^{2}-1.75\langle \tau_{f}\rangle +0.75}\right)$$
given that the tortuosity of the walk was not limited by the number of steps (black curve in Figure 3b). The observed relationship was the exponential of a rational function with a vertical and slant asymptote (Equation (20)).

The observation that the expected fiber tortuosity can be limited by the number of steps in the random walk required a second equation be defined for determining an appropriate number of steps, $N_{VMF}$. In Figure 3c, the minimum number of steps required for producing curved fibers with the expected tortuosity $\langle \tau_{f}\rangle$ is shown (black curve). The region below the black curve in Figure 3c is inaccessible, i.e., the random walk with $N_{VMF}$ steps that produces curves with $\langle \tau_{f}\rangle$ does not exist. The minimum number of VMF steps, $N_{VMF}^{*}$, required to produce curves with expectation value $\langle \tau_{f}\rangle$ is given as follows: (21)$$N_{VMF}^{*}=\left(\frac{{\langle \tau_{f}\rangle }^{2}}{2}+1\right)$$

The colored dashed curves in Figure 3c represent $N_{VMF}$ using different multiples of $N_{VMF}^{*}$ via:(22)$$N_{VMF}=\beta N_{VMF}^{*}$$
where β is the number multiplied by Equation (20) to determine the number of steps taken (Figure 3c). Using more steps than necessary, β>1, to produce curves with mean tortuosity of $\langle \tau_{f}\rangle$ was found to reduce the standard deviation of fiber tortuosity for a fixed $\kappa_{VMF}$. For instance, in Figure 3c, for β=5 the mean absolute percentage error (MAPE) between the tortuosity $\tau_{f}$ for 1000 simulated fibers and the expected tortuosity $\langle \tau_{f}\rangle$ is ~20%. When β is increased to 20, the MAPE decreases to ~20%. Therefore, as $N_{VMF}$ approaches infinity, a single value of $\kappa_{vmf}$ maps to a smaller range of possible fiber tortuosity $\tau_{f}$. If the goal of the random walk is to produce a curve whose tortuosity is very close to $\langle \tau_{f}\rangle$, then increasing β will reduce the mean absolute percentage error (MAPE) of that procedure (Figure 3c). Since, for a finite number of steps, a single value of $\kappa_{VMF}$ maps to a range of possible $\tau_{f}$, the curves can be screened for error. 

A demonstration of the simulated FM models using smoothed VMF random walks and criteria given by Equations (20)–(22) with β=20 is shown in Figure 3(d.i–f.ii), based on the experimental fiber tortuosity distributions for collagen [70], sintered metal fiber [56], and fiberboard [58]. The scattered data in Figure 3(d.i–f.i) are the experimental fiber tortuosity distributions, which are directly used to randomly select the fiber tortuosities for simulated networks (Figure 3(d.ii–f.ii)). Unlike other fiber properties, such as fiber length or diameter, a general probability density function for fiber tortuosity is elusive. Therefore, it is preferable to directly select fiber tortuosity from experimental histogram data when they are available, rather than from fitted probability densities. Since Equation (20) only defines $\kappa_{VMF}$ in terms of the expected fiber tortuosity, the process is not deterministic. To increase accuracy further, the random walks were screened. If the error in the tortuosity of a generated curve was greater than 5% of the prescribed tortuosity, the walk was discarded. Due to cubic spline smoothing, the lengths of fibers generated by the random walks were not exactly equal to $N_{VMF}\Delta_{VMF}$. Some representative FM networks are shown in Figure 3(d.ii–f.ii). Since the $\langle \tau_{f}\rangle$ for collagen is the smallest, the fibers shown in Figure 3(d.ii) are quite straight, whereas the $\langle \tau_{f}\rangle$ for fiberboard is maximized, and all of the fibers in Figure 3(f.ii) are very curvy. The extracted tortuosity distributions from the simulated FM networks are plotted as orange histograms in Figure 3(d.i–f.i), and they match quite well with the experimental data. The MAPEs between the randomly selected fiber lengths and tortuosities and the realized fiber lengths and tortuosities are reported in each plot, and they are quite small. Such a result demonstrates that using the strategy proposed in Section 2.5 we can successfully incorporate tortuosity and tortuosity distribution in FM simulation.

A set of simulations were conducted to compare the selection of VMF random walk parameters from Equations (20)–(22) with selecting them randomly. We generated 1000 fibers to reproduce the sintered metal fiber tortuosity distribution (tortuosity between 1 and 3.5). A range of $10\leq N_{VMF}\leq 100$ and $0.1\leq \kappa_{VMF}\leq 100$ was used to recreate the situation of a naïve guess and check. For comparison, Equations (20)–(22) with β=20 were used to generate a statistically identical set of fibers. Fibers were only accepted in both cases if the absolute percentage error was less than 5%. For the naïve guess and check, the process took 40 min to complete, and the rejection rate was 986 fibers per 1000 randomly generated fibers. With Equations (20)–(22) and β=20, the process was completed in 2 s, with a rejection rate of 32 fibers per 100 randomly generated fibers. This corresponds to a 1200-fold increase in speed, and Equations (20)–(22) remove the worry of guessing an appropriate range for $\kappa_{VMF}$ and $N_{VMF}$. 

### 3.3. Effect of Enforcing Non-Penetration on $\tau_{f}$, ${\hat{n}}_{f}$, and $l_{f}$ Distributions

When objects with finite size are randomly placed in a domain, such as in stochastic FM modeling, there is some probability that they overlap or penetrate one another (Figure 4(a.i,aii)). This probability increases as the volume fraction increases. For medium and long fibers, volume fractions of ~1% are sufficient to ensure that fiber overlap occurs. Such an effect may be undesirable in FM simulations. A typical method for treating fiber overlap is to treat the fibers as elastic objects that repel one another, as elaborated in Section 2.6. 

Figure 4(a.i,a.ii) show an FM model before and after enforcing non-penetration conditions. The zoomed-in figures were taken from the center of each FM model. Initially, the randomly placed fibers overlapped one another significantly (a few overlaps are circled in Figure 4(a.i)). After enforcing non-penetration conditions, the fiber overlap reduced substantially, and the fibers translated, rotated, and deformed (Figure 4(a.ii)). Figure 4b shows the change in percent fiber overlap (PFO) during the simulation. Percent fiber overlap was calculated as the total overlapping volume divided by the total fiber volume multiplied by 100. It is clear that enforcing non-penetration conditions by treating fibers as elastic bodies that repel one another is an effective means to reduce fiber penetration. However, it is also the case that this procedure can change fibers’ shapes and orientations. Therefore, the tedious work of matching fiber length, tortuosity, and orientation distributions is threatened by this procedure. Although no thorough remedy can be offered here, we statistically quantified the effects of enforcing non-penetration conditions on the fiber shape and orientation changes as well as the overall FM model.

Figure 4c shows the distribution of changes in fiber lengths, tortuosities, and orientations as a result of enforcing non-penetration conditions. The boxplots were generated from 100 fiber network simulations with mean fiber aspect ratios $\left(\langle l_{f}/d_{f}\rangle \right)$ between 10 and 150, fiber volume fractions $\phi_{f}$ between 1% and 40%, fiber orientation strengths $k_{VP}^{3D}$ between 0 and 10, and mean fiber tortuosities $\tau_{f}$ between 1.05 and 2.5, and included FM models with isotropic, transverse isotropic, orthotropic, and hexagonal symmetries. Individual fiber lengths, tortuosities, and orientations changed by ~2%, ~14%, and ~5% on average. However, the changes to the mean fiber length, tortuosity, and orientation distributions were smaller (Figure 4c): ~1%, ~9%, and ~3%, respectively. While some fibers became longer, others became shorter; while some fibers became straighter, others became curvier; and while some fibers rotated clockwise, others rotated counterclockwise. The data provided here show that enforcing non-penetration conditions has a significant effect on the final structure of the FM models—most notably on fiber tortuosity and orientation. These results suggest that more work is needed to develop better criteria and methods for enforcing non-penetration conditions—particularly criteria and methods aimed at preserving fiber tortuosity and orientation. Until then, these data serve as a rough estimate for the unavoidable errors generated in stochastic FM modeling when enforcing non-penetration conditions.

This section is intended to provide actions that may be taken to reduce the errors (for fiber length, orientation, and tortuosity) that occur when employing non-penetration conditions. However, the complexities of this issue are extreme enough to warrant a dedicated study of the topic, and no quantitative relationships can be confidently offered here. That being said, the magnitude of errors generated by enforcing non-penetration conditions is significant, as briefly discussed above.

### 3.4. Pore size Distribution Analysis of FM Models

FMs are inherently porous, and the pore structure is a direct consequence of the FM structure. Pore size $d_{p}$ is known to increase linearly with increasing fiber diameter $d_{f}$ and to decrease with the square root of the volume fraction $\phi_{f}$ [66,67,71,72,73,74,75]. A final step in verifying the structure of an FM model is to quantify the pore structure and compare it to empirical trends. The mean pore size $\langle d_{p}\rangle$ of 850 FM models was measured. The models had volume fractions $\phi_{f}$ between 0.01 and 0.65, fiber diameters $d_{f}$ from 1.5 nm to 4 mm, fiber lengths $l_{f}$ from 25 nm to 400 mm, fiber tortuosities $\tau_{f}$ from 1.03 to 6, and fiber orientation strengths $k_{VP}^{3D}$ from 0 to 20.

Univariate (Figure 5a–c) and multivariate (Figure 5d–f) analyses were conducted using traditional linear regression to relate $\langle d_{p}\rangle$ or pore size distributions to $d_{f}$, $l_{f}$, $\phi_{f}$, $\tau_{f}$, and $k_{VP}^{3D}$. Figure 5a shows the empirically consistent result with a linear relationship $\langle d_{p}\rangle \;\propto d_{f}$ [65,66,67,68,71,72,73,74], while Figure 5b shows the empirically consistent result where $\langle d_{p}\rangle \;\propto \frac{1}{\sqrt{\phi_{f}}}$ [65,66,67,68,71,72,73,74]. The data show little-to-no relationship of $\langle d_{p}\rangle$ with $l_{f}$, $\tau_{f}$, and $k_{VP}^{3D}$. As shown in Figure 5c, $\langle d_{p}\rangle$ appears to decrease linearly with $\langle \tau_{f}\rangle$. The effect is small compared to the relationships of $\langle d_{p}\rangle \;\mathrm{vs}.\;d_{f}$ and $\langle d_{p}\rangle \;\mathrm{vs}.\;\phi_{f},$ especially over typical ranges of fiber tortuosity (~1–2).

Multivariate analysis would be more insightful. Figure 5(d.i) shows that the predominant predictive variable of mean pore diameter is mean fiber diameter, with the following relationship:(23)$$\langle d_{p}\rangle \approx 2\langle d_{f}\rangle$$

Knowing the mean fiber diameter, the mean pore size could be estimated within ~60% error (Figure 5(d.ii)) using Equation (23).

Volume fraction is the second most significant predictor, as shown in Figure 5(c.i), with a semi-log plot. The predicted relationship is as follows:(24)$$\langle d_{p}\rangle \approx \frac{\langle d_{f}\rangle }{\sqrt{\phi_{f}}}$$

Knowing the mean fiber diameter and volume fraction of the FM allows for estimations of mean pore diameter within ~40% error (Figure 5(e.i,e.ii)) using Equation (24). Lastly, knowing the mean fiber diameter, tortuosity, and volume fraction, the mean pore diameter could be estimated within ~20% error (Figure 5(fi–e.ii)) using Equation (25):
(25)$$\langle d_{p}\rangle \approx \langle d_{f}\rangle \left(\frac{1}{\sqrt{\phi_{f}}}-0.05\langle \tau_{f}\rangle \right)$$

Reasonably speaking, since very high fiber tortuosities were included in the dataset, the error in predicting the mean pore diameter when knowing only the mean fiber diameter and volume fraction would likely be less than what reported here in practice. On the other hand, Equation (23) may be inaccurate when predicting mean pore diameter for very low- or high-volume-fraction FMs. Equations (23) and (24) report well-known pore diameter relationships and support the validity of stochastic FM modeling. However, Equation (25) suggests that experimental investigation of the effect of fiber tortuosity on pore structure might be warranted.

### 3.5. Limitations

In Section 3.1, a relationship for correcting de la Vallee Poussin orientation probability density parameters is presented. It shows that the prescribed corrections are useful for transverse isotropic, orthotropic, and hexagonal fiber orientation symmetries. However, the prescribed corrections are not useful for asymmetric fiber orientations. Furthermore, unless multiple measurements are taken, one must assume that due diligence has been taken to approximately align the FM’s primary fiber orientation directions with the imaging plane. In Section 3.2, relationships are provided to select parameters for a VMF random walk to produce curves with the desired tortuosity. The relationships perform well for their intended purpose; however, random walks are not deterministic; therefore, the tortuosity of a random walk cannot be predicted deterministically. The provided relationships help in selecting parameters that correspond to a high probability of producing a random walk with a desired tortuosity. Therefore, screening of random walks is still advised. In Section 3.3, the magnitude of errors that arise from enforcing non-penetration conditions is quantified. This section shows the limitations of stochastic FM modeling in cases where fiber overlap is not permissible in an FM model.

## 4. Conclusions

Stochastic modeling is a useful tool for studying FMs with numerical methods. This report addresses three critical issues in stochastic FM modeling methods, namely, projection distortion in fiber orientations, fiber tortuosity, and non-penetration conditions. Analysis of fiber orientation distributions revealed that 2D histograms of fiber orientation distributions suffer from projection distortion. This results in mismatches between the fitted orientation distributions and reality. We provided a relationship between the 2D and 3D de la Vallee Poussin probability distribution functions to correct for projection distortion. The relationship was valid and accurate for transverse isotropic, orthotropic, and hexagonal material symmetries, but inaccurate for anisotropic distributions. Further work is needed to extend the relationships to arbitrary material symmetries. After analyzing the tortuosity of paths generated by VMF random walks, we were able to develop criteria to relate the VMF random walk parameters to the expected tortuosity of the path. It is known that enforcing non-penetration conditions results in altering the shape and orientation of fibers in stochastic FM models. The extent to which fiber lengths, tortuosities, and orientations change has not been studied and reported in the literature. Here, we showed that fiber lengths changed by ~2% on average, fiber orientations changed by ~5% on average, and fiber tortuosities changed by ~14% on average, while the mean fiber length changed by ~1% on average, the mean fiber orientation strength changed by ~3% on average, and the mean fiber tortuosity changed by ~9% on average. While the change in fiber lengths was quite low, the change in fiber tortuosities and orientations was very significant for the cases studied here. Further work is needed to develop methods and criteria for enforcing non-penetration conditions, which better preserve fiber tortuosities and orientations. Lastly, an analysis of the pore structures produced by these procedures was conducted. The analysis showed that the mean pore size of these models scaled linearly with fiber diameter and was inversely proportional to the square root of the volume fraction. These results are consistent with known empirical trends. Furthermore, the analysis showed that fiber orientations and lengths had little-to-no effect on the mean pore diameter, but fiber tortuosity showed a meaningful effect. Specifically, large fiber tortuosities (greater than 2) significantly changed the mean pore diameter. Since fibers in real FMs are typically not that large, this effect is functionally negligible. However, this result may warrant experimental investigation. Further work is necessary to develop a relationship for correcting orientation distribution parameters in cases where fiber orientations are asymmetric and for predicting errors that occur when enforcing non-penetration conditions so that they can be dealt with preemptively.

## Figures and Tables

**Figure 1 polymers-14-04437-f001:**
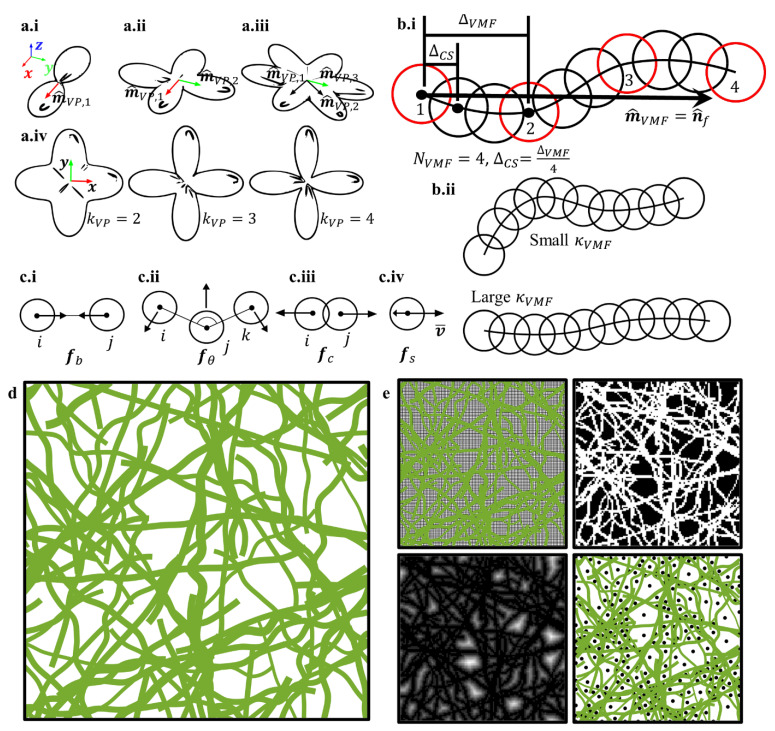
(**a.i**–**a.iv**) Schematic of the de la Vallee Poussin distribution for (**a.i**) transverse isotropic, (**a.ii**) orthotropic, (**a.iii**) and hexagonal symmetries; (**a.iv**) shows how increasing the orientation strength parameter $k_{VP}^{3D}$ affects the shape of the distribution. (**b.i**,**b.ii**) Schematic detailing the smoothed VMF random walk procedure and showing examples of low- and high-tortuosity curves: (**b.i**) A fiber is generated from a smoothed VMF random walk with step size $\Delta_{VMF}$, spline step size $\Delta_{CS}$, and preferred direction ${\hat{m}}_{VMF}={\hat{n}}_{f}$; (**b.ii**) for a low orientation strength $\kappa_{VMF}$, the simulated fibers are highly tortuous, while for a high orientation strength $\kappa_{VMF}$, the fibers are less tortuous. (**c.1**–**c.iv**) Schematic of the forces defined for enforcing non-penetration conditions: (**c.i**) bond force; (**c.ii**) angle force; (**c.iii**) contact force; (**c.iv**) drag force. (**d**) A graphic depicting the final result of the stochastic FM modeling procedures. (**e**) A 2D schematic detailing the calculation of mean pore diameter $\langle d_{p}\rangle$ using the watershed method.

**Figure 2 polymers-14-04437-f002:**
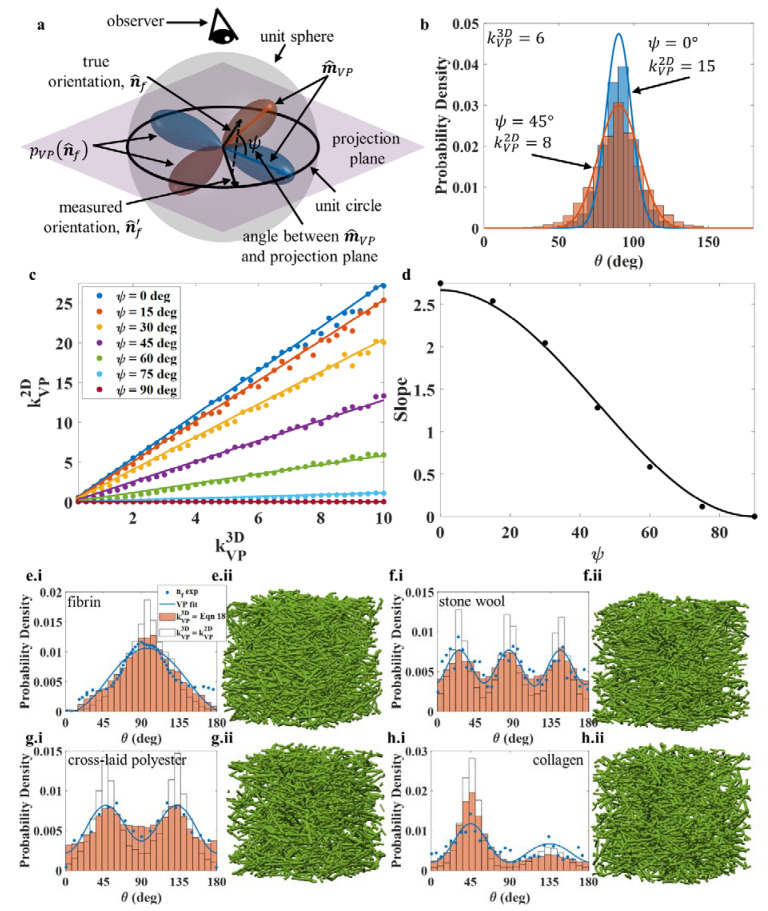
(**a**) Schematic showing two VP distribution functions with identical orientation strength but different rotations with respect to an observer. The schematic also depicts an orientation drawn from either distribution being projected onto a plane from the observer’s perspective. (**b**) Orientation distributions of both VP distribution functions in panel (**a**) after projection and binning. (**c**) The effects of $k_{VP}^{3D}$ and ψ on the measured orientation strength $k_{VP}^{2D}$. (**d**) The slope of $k_{VP}^{2D}$ vs. $k_{VP}^{3D}$ changes with the angle ψ in subfigure (**c**). (**ei**–**hii**) Curve fitting of empirical histograms and fiber orientation histograms from modeling with and without correcting for projection distortion for (**e.i**,**e.ii**) fibrin network, (**f.i**,**f.ii**) stone wool, (**g.i**,**g.ii**) cross-laid polyester, and (**h.i**,**h.ii**) collagen. $n_{f}^{exp}$ represents the experimental fiber orientation data; “VP fit” is the regression of Equation (7) to experimental data; the orange histogram is the fiber orientation distribution of simulated FM structures when projection distortion is accounted for; the white histogram is the fiber orientation distribution of simulated FM structures when directly using $k_{VP}^{2D}$ (from regression) as the orientation strength parameter $k_{VP}^{3D}$ (for drawing random fiber orientations).

**Figure 3 polymers-14-04437-f003:**
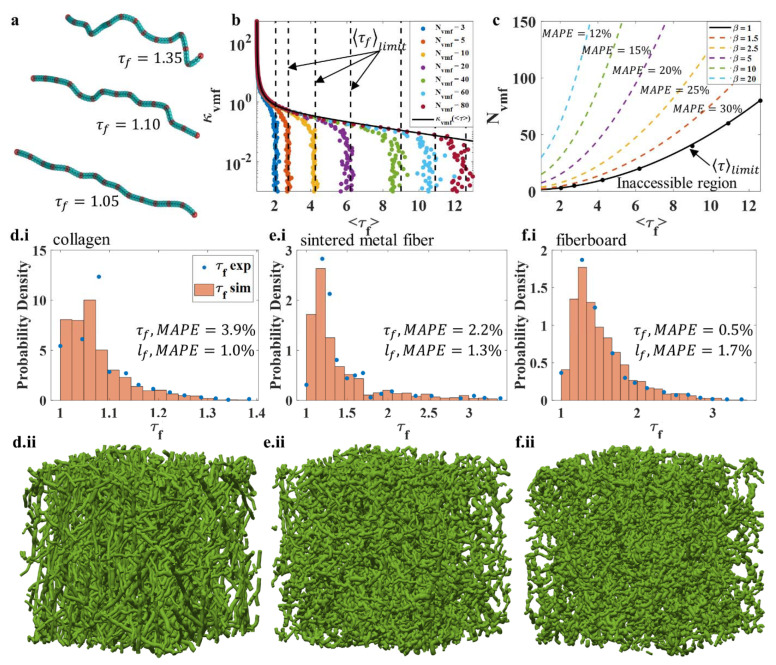
Controlling the tortuosity of VMF random walks: (**a**) Three randomly generated curves with varying tortuosity. Red spheres denote a step from the random walk, while blue spheres are points with added cubic splines. (**b**) The relationship between expected fiber tortuosity and orientation strength $\kappa_{VMF}$. (**c**) How the number of steps in the random walk affects the possible fiber tortuosity and the mean absolute percentage error between the resulting tortuosity and the expected tortuosity. (**d.i**–**f.ii**) Empirical fiber tortuosity distributions modelled using the VMF random walk and criteria for selecting $\kappa_{VMF}$ and $N_{VMF}\beta =20$ (β=20 ) for (**d.i**,**d.ii**) collagen, (**ei**,**e.ii**) sintered metal fiber, and (**f.i**,**f.ii**) fiberboard. $\tau_{f}^{exp}$ represents the experimental data for the FMs collagen [70], sintered metal fiber [56], and fiberboard [58]; $\tau_{f}^{sim}$ is the measured fiber tortuosity distribution from the simulated FMs.

**Figure 4 polymers-14-04437-f004:**
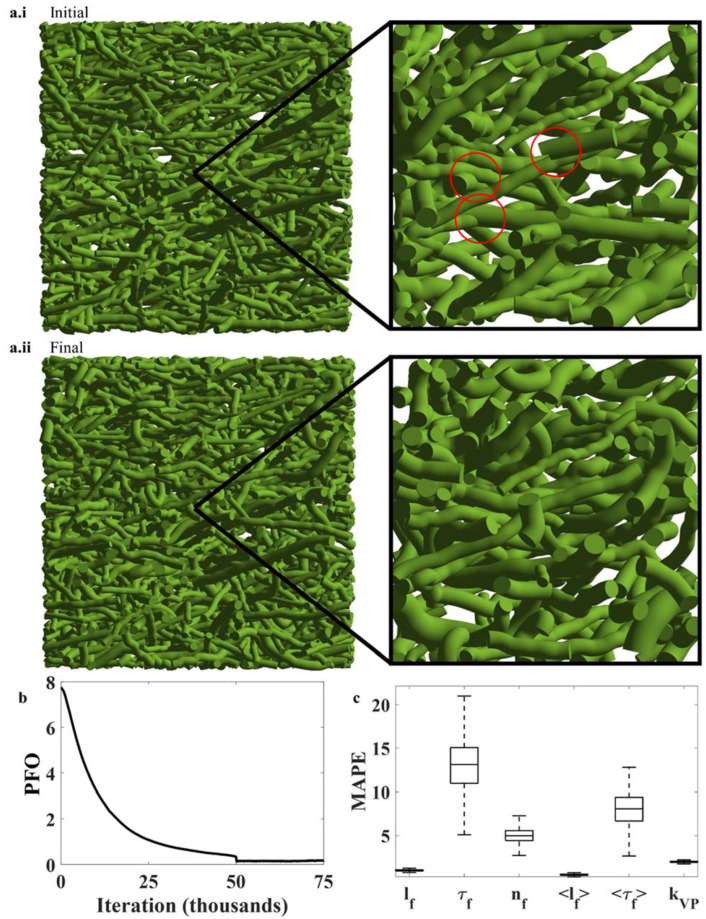
The effects of enforcing non-penetration conditions: (**a.i**,**a.ii**) An FM model before and after enforcing non-penetration conditions. (**b**) Percent fiber overlap (PFO) during the simulation. (**c**) Statistical characterization of mean absolute percentage error between the initial and final fiber lengths, tortuosities, and orientations.

**Figure 5 polymers-14-04437-f005:**
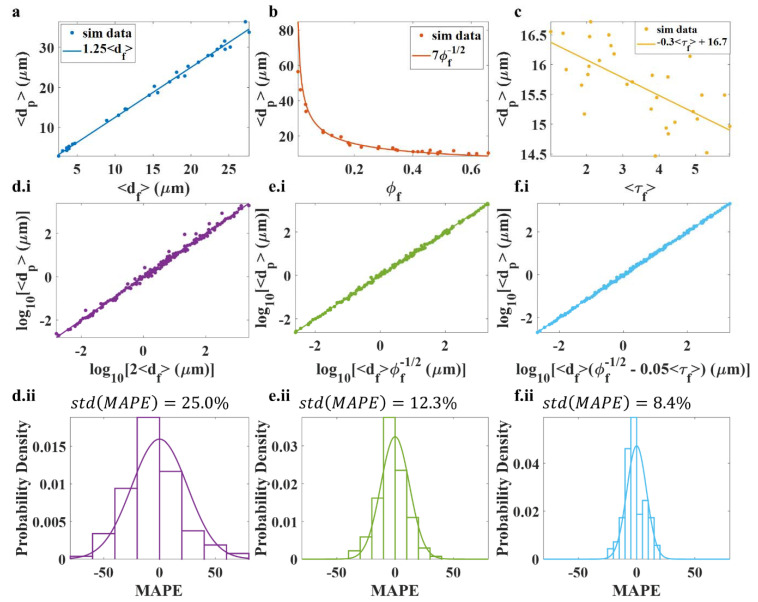
Characterization of mean pore size vs. FM model structure: (**a**–**c**) Univariate analysis of mean pore diameter vs. mean fiber diameter, fiber volume fraction, and mean fiber tortuosity. (**d.i**–**e.ii**) Multivariate analyses of mean pore diameter. (**d.i**,**d.ii**) When all properties of the FM are variable, fiber diameter is still a strong predictor of mean pore size. (**e.i**,**e.ii**) Mean fiber diameter combined with volume fraction offers better predictive power than mean fiber diameter alone. (**f.i**,**f.ii**) Mean fiber tortuosity significantly increased the predictive power for this multivariate dataset; however, this dataset is not representative of typical FMs, since it includes a very large range of fiber tortuosities. (**d.ii**–**f.ii)** show the error between the measured mean pore diameter and the predicted values.

**Table 1 polymers-14-04437-t001:** Fitted parameters for experimental fiber orientation distributions and corrected 3D orientation strength parameters.

	Fibrin	Stone wool	Polyester	Collagen
$N_{VP}$	1	3	2	2
$k_{VP,i}^{2D}$ (Measured)	0.98	5.25, 5.25, 5.25	2.50, 2.50	4.00, 2.20
$k_{VP,i}^{3D}$ (Equation (18))	0.37	3.41, 3.41, 3.41	1.33, 1.33	2.12, 1.17

## Data Availability

The data that support the findings of this study are available from the corresponding authors upon reasonable request.

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
