# Peer review of "A Tractable, Transferable, and Empirically Consistent Fibrous Biomaterial Model"

_polymers, 2022, doi:10.3390/polym14204437_

Round 1
Reviewer 1 Report
Summary
This paper deals with the stochastic modelling of fibrous material (FM). In this kind of modelling, individual fibers are generated sequentially with parameters drawn from various probability distributions (length, diameter, orientation, random walk direction). The aim of the paper is to provide general relationships and methods to determine probability distribution parameters one should use to build a 3D FM with particular properties (fibers mean length, fibers mean diameter, fibers orientation distribution, fibers tortuosity distribution), extracted from experimental measurements. The question of how enforcing non-penetrating condition between fibers affects those properties is also examined.
These issues are essentially tackled by generating large dataset of 3D stochastic FMs, and extracting empirical rules between model input parameters and resulting FM properties. This systematic investigation is at the core of the paper and at the end of the day, the authors highlight a set of expressions and methods that are useful for the community to generate 3D stochastic FMs and investigate structure-property relationship in this class of material.
General comments
The problems addressed in this study are definitely of interest, and this is well presented in the introduction. Overall, the paper is well written and the methodology is clearly explained. Unfortunately, no analytical solutions to the various questions covered in the paper are provided, but this is likely due to their complexity and the systematic investigations done by the authors appear as a reasonable and useful workaround. The main results from this work are clearly of interest for the community.
Yet it is sometimes a bit unclear what is new or not in the paper. In particular, the method section is a very clear and detailed presentation of the stochastic model used in this study, but it is not clear if the model, or part of it, or the combination of the various methods, is original or not. Could the authors make that clearer in the text? In the same vein, the novelty of section 3.4 dedicated to pore size distribution is not clear. No one has ever tested the relationship between pore size distribution and stochastic model parameters before? Is it sometimes an issue with other approaches?
Finally, regarding section 3.3, effect of enforcing non-penetrating conditions between fibers. I understand that this issue is difficult to handle, and I appreciate that the authors, even if they could not find any direct solutions, try to evaluate the impact of their molecular mechanics relaxation on the FMs properties. However, I think that it is a bit of a shame that they report only variations of mean values on their whole set of data. Would it be possible to highlight more precise relationships, e.g. when the mean fiber length is in this range the tortuosity increases or decreases like that when non-penetrating conditions are enforced and so on… And if there is no correlation between any variables, I think it would be worth mentioning.
Specific comments
1) In Equation 1, isn't there a in the gamma distribution denominator?
2) Equations 6 and 7 and related paragraph. As I understand, for 2D distributions, which result from the projection of experimental 3D distributions, we can set because all projected fibers lie in the same plane. Yet I do not get how expression 6 and 7 were derived. Can you provide a derivation of these equations, or a reference to it? In particular, I could not figure out where the gamma function comes from in the expressions. Also, since we set , does it mean that when constructing a 3D distribution from a 2D distribution we always assume that (in 3D) all fibers lie in parallel planes, i.e. vectors are always orthogonal to a same plane ( ? If so that should be highlighted because this is quite a strong limitation in my opinion, and if on the contrary my understanding is wrong, could you please try to clarify this in the text?
3) Equation 9, aren't there missing squares in the denominator?
4) Equation 19. This is really a tiny detail, but would it be worth defining as the integral between 0 and 2π instead of 0 and π? Because experimentally I don't think we can choose to probe angles specifically in the range [0, π] to calculate the average, since the sample orientation is precisely the unknown.
5) End of page 9, beginning of page 10, discussion around the orientation distributions of 4 different reconstructed FMs. For collagen, there is a discrepancy between projected 3D distributions and expected 2D distributions, i.e. relation 18 breaks apart. This is attributed to the fact that the 2D distribution is not symmetric. But is it also link to comment #2? Also, have you tried other projection angles than ? Maybe that would help to reconcile 3D and 2D distributions? Finally, I guess that generating 3D FMs and projecting fibers orientations on 2D histograms is quite fast computationally speaking. Then, for instance on collagen, do you think that it would be possible to design an algorithm that would start with a random orientations distribution and progressively alter this distribution until calculated and experimental 2D histograms match?
6) Right after equation 22, "beta is the number multiplied by equation 20 to determine the number of steps taken". I could not understand this statement. In addition, in the following of the section the authors argue that "the extracted tortuosity distributions from the simulated FM networks are plotted as orange histograms in Figures 3di-3f.i, and they match quite well with the experimental data". Well, since in the procedure the author discard generated fibers with a tortuosity that deviates more than 5% of the prescribed value, I guess I could have done the same with any random algorithms that generate curvy fibers and obtain a good agreement with experimental data as well. Therefore, that would also be interesting to see the histograms obtained without discarding fibers that deviate from the prescribed tortuosity (but definitely keep this discarding procedure for the rest).
7) Figure 4b, y axis label should be PFO.
8) Figure 4c, on the x label there are and . I guess that should be and .
9) Last sentence page 24, figure 5ci should be figure 5ei.
Author Response
See attached PDF file "Reply to reviewers".

Reviewer 2 Report
1. Email from all of the authors needs to be included based on MDPI format.
2. Quantitative results need to be added in the abstract section.
3. As your abstract's final sentence, include a "take-home" message.
4. Keywords should have been reorganized alphabetically.
5. The novel of the present study is not clear. Many published literature has been widely studied in the past. Further explanation in the introduction section in advance is mandatory.
6. The work, novelty, and limitations of similar prior studies must be explained in the introduction section to highlight the research gaps that the current study aims to fill.
7. The end of a paragraph in the introduction section should explain the objective of the present article, the present form was not.
8. The authors need to explain the advantage of computational study (in silico) as conducted in the present study compared to experimental study (in vitro) and clinical study (in vivo), such as faster results and reduces cost. It is a crucial issue that authors should provide in the introduction and/or discussion section. The MDPI's suggested reverence should be adopted as follows: Jamari, J.; Ammarullah, M. I.; Santoso, G.; Sugiharto, S.; Supriyono, T.; Heide, E. van der. In Silico Contact Pressure of Metal-on-Metal Total Hip Implant with Different Materials Subjected to Gait Loading. Metals (Basel). 2022, 12, 1241. https://doi.org/10.3390/met12081241
9. To help the reader grasp the study's workflow more easily, the authors could include more visuals to the materials and methods section in the form of figures rather than sticking with the text that now predominates.
10. A comparative assessment with similar previous research is required.
11. Before moving on to the conclusion section, the present study's limitations must be included.
12. Conclusion section is missing, provide it.
13. Please explain the further research in the conclusion section.
14. The authors recommended including an additional reference from five years ago. MDPI-published literature is encouraged.
15. Due to grammatical and language issues, the authors need to proofread the present work. This problem would use MDPI English editing service.
16. Please ensure that the authors followed the MDPI format correctly; modify the current form and recheck, as well as any other problems that have been highlighted.
Author Response
Please see the attached PDF file "Reply to reviewers".

Round 2
Reviewer 2 Report
Reviewers greatly appreciate the efforts that have been made by the author to improve the quality of their articles after peer review. I reread the author's manuscript and further reviewed the changes made along with the responses from previous reviewers' comments. Unfortunately, the authors failed to make some of the substantial improvements they should have made making this article not of decent quality with biased, not cutting-edge updates on the research topic outlined. In addition, the author also failed to address the previous reviewer's comments, especially on comments number 5, 6, and 8. With all due respect, the reviewer opposed this article to be published and must be rejected. Thank you very much for the opportunity to read the author's current work.
Author Response
Please see attached PDF 'Reply to reviewer #2'. Thanks.

Round 3
Reviewer 2 Report
The reviewer's investigation results show that this article is still not worthy of acceptance due to its minimal contribution. Please the author to read carefully the previous comments and make improvements according to the results of the review. Still, with the previous decision, the reviewer opposed this article for publication and should be rejected. Thank you for the opportunity to review this article and I appreciate the author for his efforts.